# How Does Data Diversity Shape The Weight Landscape of Neural Networks?

## Abstract

To enhance the generalization of machine learning models to unseen data, techniques such as dropout, weight decay ($L_2$ regularization), and noise augmentation are commonly employed. While regularization methods (i.e., dropout and weight decay) are geared toward adjusting model parameters to prevent overfitting, data augmentation increases the diversity of the input training set, a method purported to improve accuracy and calibration error. In this paper, we investigate the impact of each of these techniques on the parameter space of neural networks, with the goal of understanding how they alter the weight landscape in transfer learning scenarios. To accomplish this, we employ Random Matrix Theory to analyze the eigenvalue distributions of pre-trained models, fine-tuned using these techniques but using different levels of data diversity, for the same downstream tasks. We observe that diverse data influences the weight landscape in a similar fashion as dropout. Additionally, we compare commonly used data augmentation methods with synthetic data created by generative models. We conclude that synthetic data can bring more diversity into real input data, resulting in a better performance on out-of-distribution test instances.

## 1 Introduction

Machine learning (ML) models excel at discovering patterns from training data, which allows them to make predictions on unseen data. However, as models grow more complex, they risk a phenomenon called *overfitting* – the tendency of models to capture not only meaningful patterns but also noise and random variations from the training data, resulting in poor generalization to new instances. To address this, model regularization techniques, such as *dropout* (Srivastava et al., 2014) and *weight decay* ($L_2$ regularization)(Krogh & Hertz, 1991), are used in conjunction with Deep Neural Networks (DNNs) to enhance their generalization capability. While *dropout* reduces model complexity by randomly deactivating neurons during training, *weight decay* adds a penalty to the loss function based on the magnitude of model weights. These techniques shape the model's weight landscape in different ways: dropout works by randomly deactivating neurons during each training iteration, forcing the network to distribute learning across multiple independent pathways rather than over-relying on particular features. Weight decay, on the other hand, encourages a smoother and more evenly distributed weight structure by penalizing large weights, pushing the model toward simpler and more generalizable solutions (Andriushchenko et al., 2023; Zhang et al., 2018).

Beyond direct parameter adjustments, empirical regularization techniques such as adding noise to input data or employing data augmentation have also proven effective for improving model generalization (Bishop, 1995). By exposing models to greater breadths of input patterns, these methods help improve their adaptability to diverse, real-world data. Despite existing evidence of their effectiveness, the underlying mechanisms of data augmentation remain an insufficiently studied area of research. Recently, synthetic data generated by models such as stable diffusion (Rombach et al., 2022) and large language models (LLMs) (Brown, 2020), has emerged as a promising, new approach to boost model performance(Sahu et al., 2023; He et al., 2022). Synthetic data not only augments existing datasets but also simulates rare events and edge cases (Santoso et al., 2017).

Similar to dropout, data augmentation acts as a form of regularization by introducing purposeful variability into the model's training process. Techniques such as rotation, flipping, cropping, random erasing, color jittering, and mixup are used to modify input image data in deep neural networks

(DNNs), therefore increasing the diversity of the data seen during training . This broader input variation helps prevent overfitting by encouraging the model to capture more generalizable features and patterns. Dropout shares the goal of enhancing diversity but achieves it by randomly deactivating neurons, forcing the model to explore different configurations of its weights. Weight decay, on the other hand, works by penalizing large weights to simplify the model, thus controlling complexity and preventing it from fitting noise in the training data. Ultimately, these regularization strategies aim to enhance model generalization – either by promoting adaptability through diverse representations (dropout and data augmentation) or by constraining complexity (weight decay) to avoid overfitting.

Given that these regularization methods all seek to improve generalization, albeit through different mechanisms, a natural question arises: **does data augmentation have a similar effect on the neural network's weight landscape as dropout or weight decay?** To investigate our hypothesis, we utilize Random Matrix Theory (RMT), a statistical framework for analyzing the structure of large matrices, to analyze the changes in weight matrices across different regularization approaches. The weight matrix $W$ from a specific layer can be approximated as the sum of a random noise matrix $W_{\text{rand}}$ and a difference matrix $W_\Delta$, so that

$$W \approx W_{rand} + W_\Delta$$

After a layer has been properly trained, either through extensive training on labeled data or fine-tuning on a downstream task, the difference matrix $W_\Delta$ provides meaningful structural patterns that represent learned features, which are distinct from the random noise components $W_{\text{rand}}$ (Martin & Mahoney, 2019). Thus, examining $W_\Delta$ can provide insights into the effects of data diversity on the learned weight matrices (Pennington & Worah, 2017; Sagun et al., 2017), facilitating comparisons of robustness among regularization approaches. Specifically, measures of weight matrices, such as the empirical spectral density (ESD), reveal characteristic signals that indicate the presence of implicit self-regularization during training (Martin & Mahoney, 2019).

In this work, we focus on the transfer learning domain and investigate the changes in weight matrices between the pre-trained and fine-tuned models in response to data augmentation techniques. By examining these changes, we aim to understand the underlying mechanisms by which diverse input data improves model performance and generalizability. Previous studies have shown that fine-tuning can significantly alter the spectral properties of weight matrices, often resulting in changes that are consistent with improved feature specialization and adaptation to new tasks(Li et al., 2020). We investigate the effects of real and synthetic data on the weight landscape, using RMT to show how data augmentation interacts with regularization techniques such as dropout and weight decay. Additionally, we evaluate the impact of traditional and synthetic data augmentation on model robustness across both in-distribution (ID) and out-of-distribution (OOD) tasks.

**Our contributions** can be summarized as follows: (1) We investigate the changes in the weight space of Deep Neural Networks (DNNs) resulting from varying levels of data diversity; (2) we provide theoretical and empirical insights into why diverse data improves model performance, linking these findings to changes in learned feature representations and their robustness; and (3) we explore the use of synthetic data generated through generative models, evaluating its effect on both in-distribution (ID) and out-of-distribution (OOD) tasks to determine how it complements traditional data augmentation approaches in improving model robustness and generalization. Our study offers a comprehensive perspective on how data diversity, both from real and synthetic sources, influences the internal dynamics of DNNs.

## 2 METHODS

### 2.1 PRELIMINARY

This section introduces data diversity measurement and the basics of RMT, as well as the metrics that will be used in the subsequent analyses.

#### 2.1.1 REGULARIZATION

**Dropout.** Dropout (Srivastava et al., 2014) helps minimize overfitting during training by randomly disabling neurons in fully connected layers based on a given probability $p$. This randomness is

controlled by a Bernoulli distribution, where neurons are either kept or deactivated during each training iteration. However, during inference, all neurons are active. To ensure consistency between training and inference, the output is scaled by $1 - p$ during training. Dropout has become almost a default method for training or fine-tuning deep neural networks. In this paper, we implement dropout operations on different layers in transformer-based models.

**Weight Decay.** Weight decay (Krogh & Hertz, 1991; Ishii & Sato, 2018) is another commonly used regularization technique to prevent overfitting. It works by adding a small penalty, which is proportional to the size of weight parameters, to the loss function during training. This encourages the model to keep weights smaller. During training, the weights are updated not only based on the prediction error-induced losses but also on this penalty, which pushes the model to find simpler solutions.

**Data Augmentation.** Data augmentation (Rebuffi et al., 2021; Shorten & Khoshgoftaar, 2019) is a strategy to artificially expand the size and diversity of training datasets without requiring the acquisition of new data. This technique involves applying a range of transformations to existing samples, such as flipping, cropping, adjusting brightness, and adding noise. Mixup (Zhang, 2017) is a specific data augmentation technique that creates new training examples by linearly combining pairs of input data and their corresponding labels. By introducing these variations, data augmentation enables models to learn more robustly from a wider array of instances, improving generalization. This approach is particularly advantageous in domains related to image tasks.

### 2.1.2 DIVERSITY MEASURE

**Vendi Score** (Dan Friedman & Dieng, 2023) has been recently proposed for measuring diversity in ML data. This score quantifies data diversity based on the similarities between elements in a dataset, making it particularly useful for evaluating the effectiveness of data augmentation techniques and assessing the diversity of synthetic samples from generative models. The score is derived by using a set of samples along with a pairwise similarity function, which calculates a value that reflects the effective count of unique elements in the dataset. More specifically, given a positive semi-definite matrix $K \in \mathbb{R}^{n \times n}$ representing similarity scores, the Vendi Score (VS) is defined as

$$VS(K) = \exp\left(-\text{tr}\left(\frac{K}{n}\log\frac{K}{n}\right)\right) = \exp\left(-\sum_{i=1}^{n}\lambda_i\log\lambda_i\right),$$

where $\lambda_i$ are the eigenvalues of the matrix $K/n$, with the definition of $0\log 0 = 0$. In essence, VS is the exponential of the von Neumann entropy of the scaled similarity matrix, $K/n$, equivalent to the Shannon entropy of this matrix's eigenvalues, also known as effective ranks. The higher the Vendi Score, the more diverse the dataset is. For evaluating images, Vendi Score provides two correlation methods (as the similarity function) – the pixel-pairwise correlation by comparing raw pixels directly and the correlation of the embeddings learned from pre-trained models. We use VS as a diversity measure to evaluate different datasets. The default embedding model of VS is INCEPTION V3, which we replace with CLIP in this paper.

From Figure 1 one can see that commonly used image augmentation methods typically increase input data diversity. Some methods (e.g., Auto-v3) increase the diversity score at a larger rate than others. Additionally, we observe that as the number of label classes increases, the diversity score initially increases but then exhibits a drop afterward. This suggests that, to a certain extent, the output label diversity can be captured by input similarities. We also observe that pixel-wise diversity scores do not always match embedding-wise scores after applying data augmentation. Similarly, we will see later that synthetic data itself is less diverse than real data but when it is combined with real data, the combined dataset can often reach a higher diversity score.

### 2.1.3 RANDOM MATRIX THEORY

Consider a $d$-layer DNN with corresponding weight matrices $W_1, W_2, \ldots, W_d$. For each weight matrix $W_i$ with shape $N \times M$, assume without loss of generality that $N \geq M$ (otherwise, consider $W_i^\top$). The correlation matrix is defined as

$$X_i = W_i^\top W_i.$$

The eigenvalues of $X_i$ are given by $\{\lambda_j\}_{j=1}^{M}$.

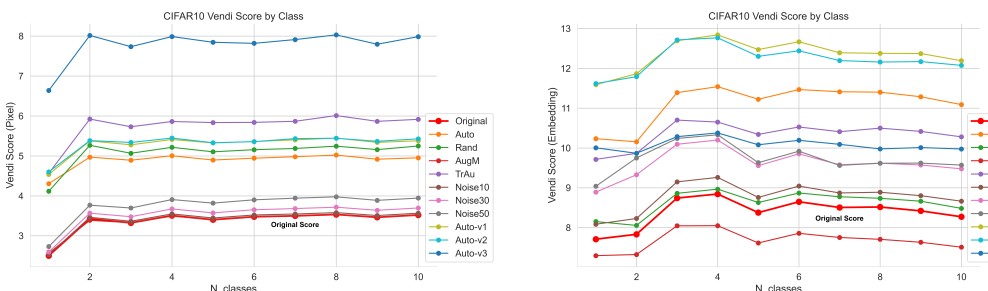

Figure 1: Data diversity on CIFAR-10 measured by Vendi Score. **Left** is the result based on pixel-wise similarity; **Right** is the result based on the similarity of embedding learned by CLIP. The red-colored line contains the VS values of the original CIFAR-10 dataset with different numbers of label classes. We employ automatic data augmentation packages to create other datasets: Auto, Rand, AugM, and TrAU. Noise10, Noise30, and Noise50 are created by adding Gaussian noise with a ratio of 0.1, 0.3, and 0.5. Auto-v1 to Auto-v3 are variants based on AutoAugment.

Let $p(x)$ denote the correlation matrix's Empirical Spectral Density (ESD). According to RMT, the ESD of a Gaussian-noise random weight matrix follows the Marchenko-Pastur (MP) distribution when the matrix dimension grows large. The MP distribution often exhibits a bulky shape. However, the empirical study of Martin and Mahoney (Martin & Mahoney, 2021) showed that, for a well-trained DNN model, specifically for a pre-trained large model, the ESD of its weight correlation matrix is heavy-tailed, not following the Gaussian noise-based MP distribution. They called it the Heavy-Tailed Self Regularization theory. Specifically, a Power Law (PL) distribution can be used to fit the heavy tail, which is given by

$$p(x) \propto x^{-\alpha}, \quad x_{\min} < x < x_{\max}. \tag{1}$$

Here, $x$ takes values in the interval $(x_{\min}, x_{\max})$, and $x_{\max}$ is chosen to be the maximum eigenvalue of the empirical correlation matrix, while $x_{\min}$ is selected to obtain a better power-law fitting, which is not generally equal to the minimum eigenvalue.

The discussion above justifies the decomposition of weight matrix $W$ into two components – a random matrix $W_{rand}$ and a difference matrix $W_{\Delta}$, which are represented by the bulk and tail of ESD, respectively, based on Martin and Mahoney's theory. Analyzing properties of both $W_{rand}$ and $W_{\Delta}$ can obtain insights into how models are trained. Figure 2 shows the $\lambda_{min}$ are selected to divide the ESD into bulk ($W_{rand}$) and tail ($W_{\Delta}$) parts. The exponent $\alpha$ of a fitted PL distribution characterizes the tail behavior of the ESD, which is in close relevance to the model's generalization capability. A smaller $\alpha$ represents a more "heavy-tailed" landscape in ESD. More large eigenvalues in the tail of ESD will exert more influence on the model's weight spaces.

## 2.2 ANALYSIS APPROACH

Due to the heavy-tailed nature of pre-trained models, we cannot obtain direct conclusions if our evaluation is based on a single metric. Instead, we focus on the trend of how regularization and diverse data influence the weight spectrum by calculating the differences in multiple metrics before and after fine-tuning on specific downstream tasks to gain valuable insights. For example, the initialized $\alpha$ in the classification layer is around 6 while its value after fine-tuning can still display a certain degree of heavy-tailed property (2-6 typically). The relative changes compared with the baseline (no regularization applied) tell us the direction and magnitude of the impact of a regularization method on the weight landscape.

Let $M$ be a set of metrics that can capture the properties of the spectrum in weight matrices. $\Delta M$ denotes the distance of these metrics in the spectral space between pre-trained models and fine-tuning models with/without regularization or data augmentation, i.e.,

$$\Delta M = M_{ft} - M_{pre}.$$

These metrics can be categorized from two perspectives – scale metrics and shape metrics. Scale metrics measure the magnitude of eigenvalues or the overall size of the weight matrix, while shape

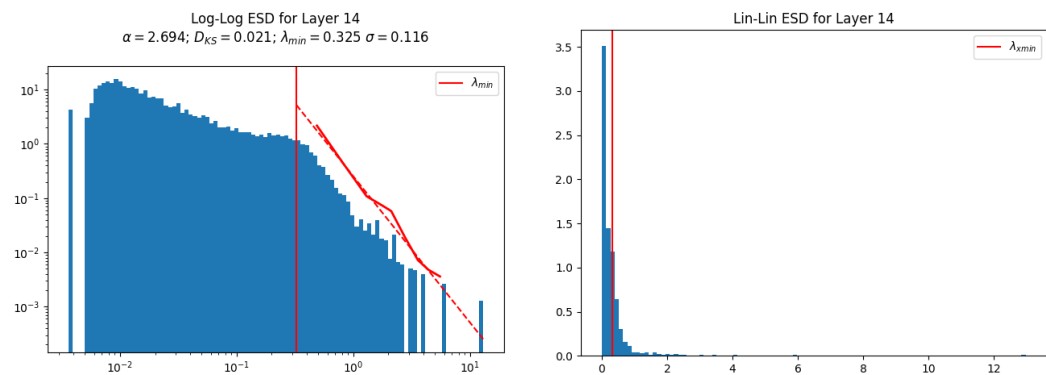

Figure 2: Empirical Spectral Density (ESD) from a feedforward layer of a fine-tuned CLIP model on CIFAR-100 with the linear (Right) and logarithm (Left) plotting scales. The bulk and tail parts are separated by the selected $\lambda_{min}$. Clearly, $\alpha$ guides the "heavy-tailed" nature.

metrics describe the distribution and structure of the eigenvalues. Scale metrics include Frobenius Norm, Spectral Norm, Stable Rank, and MP Soft Rank (Martin & Mahoney, 2019). The definitions of some of these metrics are given below:

$$\text{Frobenius Norm:} \quad \|W\|_F^2 = \|X\|_F = \sum_{i=1}^{M} \lambda_i$$

$$\text{Spectral Norm:} \quad \|W\|_\infty^2 = \|X\|_\infty = \lambda_{\max}$$

Shape metrics include power law exponent $\alpha$, matrix entropy, number of power-law spikes (*num_pl_spikes*), etc.

$$\text{Matrix Entropy:} \quad H(W) = -\sum_i \lambda_i \log(\lambda_i)$$

The metric *num_pl_spikes* counts the number of eigenvalues that are larger than $x_{\min}$.

Table 1 describes how the changes in these metrics may reflect the changes in weight space and the corresponding characteristics of its spectrum. Our spectral analysis is implemented by a tool called WeightWatcher(Martin et al., 2021), which provides multiple scale and shape metrics.

## 3 EXPERIMENT

### 3.1 EXPERIMENT SETUP

We employ the state-of-the-art text-to-image model CLIP (Radford et al., 2021) given its powerful capability of performing image classification tasks and fine tune CLIP on CIFAR-10, CIFAR-100 (Krizhevsky et al., 2009), Imagenette (Howard, 2019), Flowers102 (Nilsback & Zisserman, 2008), Stadford Cars (Krause et al., 2013), DomainNet (Peng et al., 2019). During fine-tuning, we freeze the text encoder, thus only the weights in the image encoder and the last classification layers are updated. The pre-trained vision model we used is VIT-B32 and ResNet50. The setting of the following training hyperparameters – a learning rate of 1e-5, 5 epochs, and a batch size of 32 – are kept the same in all experiments. Model performance measures are evaluated on the real test images.

### 3.2 IMPACT OF REGULARIZATION

The default CLIP-VIT-B32 model does not apply dropout in the image encoder. To investigate the impact of dropout on both hidden layers and output layers, we add dropout layers in both *Multi-headAttention* layers and *feedforward* layers within each transformer block, as well as in the final *classification* layer. In the hidden layers, dropout is applied after the *normalization* layer. For CLIP-ResNet50, we add dropout after each block. The dropout rate in the baseline model is zero, and we subsequently deploy the dropout rates of 0.1, 0.3, 0.5, and 0.7 to conduct a comparison study. The

| Metric | Change | $W$ | ESD |
|---|---|---|---|
| $\alpha$ | ↑ | Less complex $W$, fewer directions dominate. | A sharper drop-off in ESD, less "heavy-tailed" |
| | ↓ | More complexity, more dominant directions. | A slower decay in ESD, more eigenvalues with large magnitudes |
| Matrix Entropy | ↑ | More diverse and less structured weight patterns | More uniform distribution of $\lambda$ across the spectrum |
| | ↓ | More organized and meaningful weight structures | More concentrated around a few large $\lambda$ |
| Frobenius Norm | ↑ | Larger weight values overall | Larger $\lambda$ in total |
| | ↓ | Smaller weight values | Overall $\lambda$ shrink |
| Spectral Norm | ↑ | At least one dominant direction in $W$ | More outliers in the long tail |
| | ↓ | The maximum influence of any single direction is reduced | Smaller maximum $\lambda$, more compact distribution |
| num_pl_spikes | ↑ | More important and distinct features in $W$ | More $\lambda$ separate from the $W_{rand}$ |
| | ↓ | Weight structure becomes more random-like | More uniform bulk distribution with fewer outliers |

Table 1: Interpretations of metric changes on the weight matrix and its spectral density

weight decay values are varied with six levels – 1e-5, 5e-5, 1e-4, 5e-4, 1e-3, and 5e-3 – to investigate how they change weight matrices.

Figure 3 shows the scale-shape plot of distance measure ($\Delta M$) on CIFAR-10/100, Imagenette, Flower102, StandfordCars, DomainNet(Real) datasets. The horizontal and vertical dash lines show the position of the baseline, to which no regularization technique is applied. These lines divide the plot into 4 quadrants, which assist in understanding the impact of a regularization technique on the weight matrix as comparing to the baseline. If two different regularization techniques give similar change directions of $\Delta M$, we consider they have similar impacts on the weight space. Compared with the baseline, on multiple test datasets, dropout tends to increase the value of $\alpha$, reduce the Frobenius Norm, and increase the matrix entropy with an increasing dropout rate. In particular, dropouts tend to reduce the entire size of the weight matrix. On the other hand, increasing weight decay leads to a drop of $\alpha$ significantly, and it decreases the matrix entropy, which causes a more heavy-tailed ESD. Similar to dropout, weight decay reduces the size of the weight matrix. Smaller weight decay values make the overall size of the weight matrix shrink more effectively.

**Summary.** After fine-tuning the same dataset, dropout, and weight decay display a certain degree of resembling behaviors on some ESD metrics but diverge on others. In scale metrics, both dropout and weight decay can significantly shrink the size of the weight matrix and diminish the largest eigenvalue. But in shape metrics, dropout encourages a more evenly distributed ESD, while weight decay tends to make long-tailed ESD, thus increasing the nonrandom part of the weight matrix.

### 3.3 IMPACT OF DATA DIVERSITY

To increase training data diversity, we implement ten types of data augmentation: First, four automatic image augmentation techniques in PyTorch are applied: *AutoAugment* (Cubuk et al., 2018), *RandAugment* (Cubuk et al., 2020), *AugMix* (Hendrycks et al., 2019), and *TrivialAugment*(Müller & Hutter, 2021). Then we add *Gaussian noise* into the original images with a ratio of 0.1, 0.3, and 0.5. Finally, we created three advanced augmented datasets by combining *AutoAugment* with other data augmentation methods. They are *auto-v1*: AutoAugment + Gaussian noise with 0.5 ratio; *auto-v2*: AutoAugment + Gaussian noise with 0.5 ratio + RandomHorizontalFlip + RandomVerticalFlip;

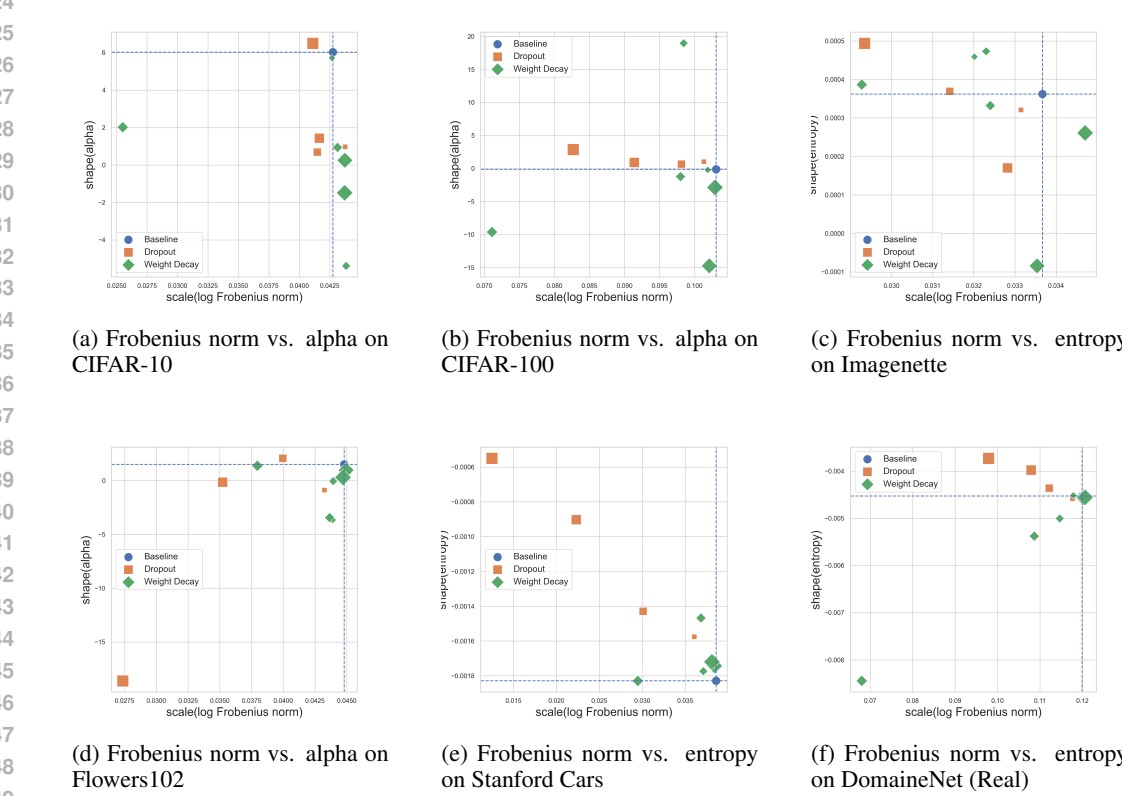

(a) Frobenius norm vs. alpha on CIFAR-10

(b) Frobenius norm vs. alpha on CIFAR-100

(c) Frobenius norm vs. entropy on Imagenette

(d) Frobenius norm vs. alpha on Flowers102

(e) Frobenius norm vs. entropy on Stanford Cars

(f) Frobenius norm vs. entropy on DomaineNet (Real)

Figure 3: Various scale-shape plots of ESDs of the classifier layers on CIFAR-10/100, Imagenette, Flower102, StandfordCars, DomainNet(Real). The results are marked with square and diamond shapes for dropout and weight decay regularization techniques, respectively. The size of markers represents the magnitude of the dropout rate and weight decay values. The results above are based on the VIT-32 backbone. Please check Figure 6 in the Appendix for results of the ResNet50 backbone.

*auto-v3*: AutoAugment + Gaussian noise with 0.5 ratio + ElasticTransform. We try to combine multiple data augmentations to explore how diverse datasets we can create (also see Figure 1).

Figure 4 adds the triangle markers representing multiple data augmentation methods into scale-shape plots. They illustrate how training data diversity can alter the spectral distributions of weight matrices. Comparing them with dropout and weight decay, the effects of data diversity are more aligned with dropout than weight decay. In both scale and shape metrics, from Figure 4 (b)(d)(e), we observe diverse data can reduce the Frobenius norm and increase the matrix entropy, thus shrinking the size of weight matrix in the same fashion of dropout and weight decay. Moreover, its change magnitude is comparable with that of dropout. Figure 4(a)(f) also shows the behavior of the entropy due to data diversity is similar to that of dropout. Overall, more resembling behaviors of shape metrics have been seen in the pair of data diversity and dropout than in the pair of data diversity and weight decay. Readers can find more scale-shape plots about other layers in Appendeix.

**Summary.** Based on spectral analysis, we find many similarities between data augmentation and dropout. By fine-tuning CIFAR-10/100, Imagenette, Flower102, StandfordCars, DomainNet(Real) datasets, they display many common tendencies of impact on the weight matrix when compared with the baseline. Data augmentation increases input data diversity, thus enriching features in the input data, while dropout forces the neural networks to reduce the reliance on any specific neurons. The induced effects of these methods on weight matrices seem to agree with each other – they reduce the overall size of the weight matrix, thus limiting the maximum influence of any single weight correlation direction and making the weight matrix less complex. Moreover, they have a similar impact on shape metrics such as matrix entropy. However, diverse data sometimes has opposite effects on shape metrics as compared with those of weight decay.

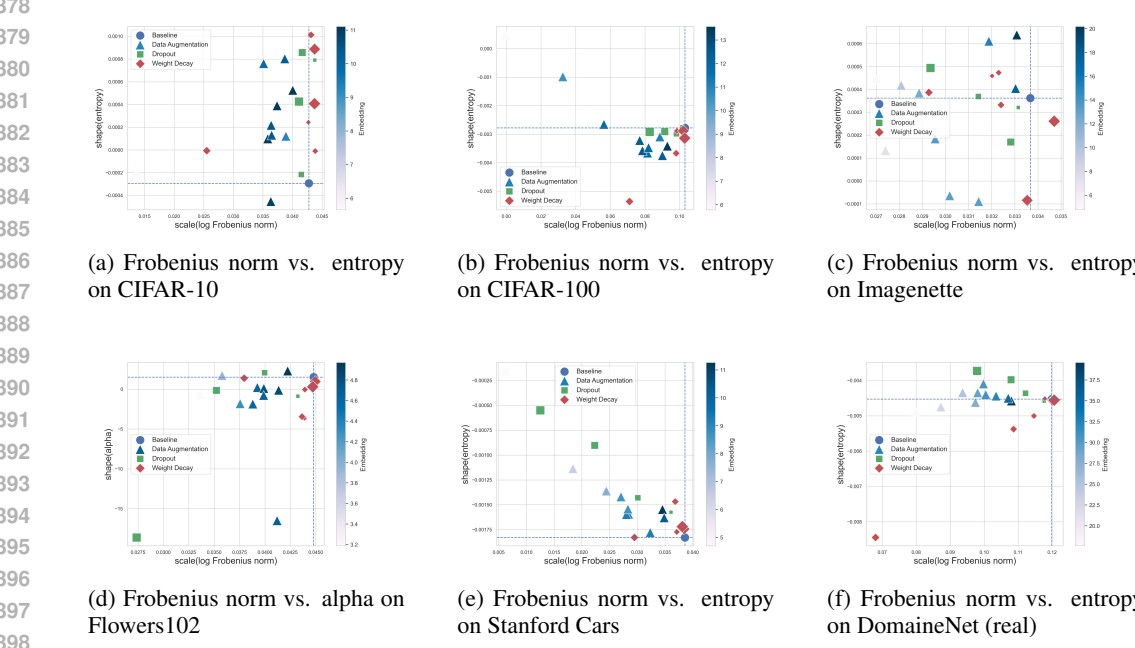

(a) Frobenius norm vs. entropy on CIFAR-10

(b) Frobenius norm vs. entropy on CIFAR-100

(c) Frobenius norm vs. entropy on Imagenette

(d) Frobenius norm vs. alpha on Flowers102

(e) Frobenius norm vs. entropy on Stanford Cars

(f) Frobenius norm vs. entropy on DomaineNet (real)

Figure 4: Various scale-shape plots of ESDs of the classifier layers on CIFAR-10/100, Imagenette, Flower102, StandfordCars, DomainNet(Real). The results are marked with triangle, square, and diamond shapes for data augmentation, dropout, and weight decay, respectively. The color intensity of triangles represents the magnitude of embedding-based diversity scores. The results above are based on the VIT-32 backbone. Please check Figure 7 in the Appendix for results of the ResNet50 backbone.

## 3.4 DIVERSITY OF SYNTHETIC DATA

To investigate whether or not synthetic data can bring in more input data diversity and thus further enhance model performance in the transfer learning context, we utilize the state-of-the-art image generative model *Stable Diffusion v1-5* (SD) (Rombach et al., 2022) to generate synthetic images. We use Domainnet (Peng et al., 2019) datasets to evaluate the CLIP model performance on both in-distribution (ID) and out-of-distribution (OOD) tasks. Specifically, for ID tasks, we use 50000 instances from the "real" category (i.e., they are real images, as opposed to artificially stylized images), which covers 142 image labels. The baseline model is trained on these instances and evaluated on "real" test images. Data augmentation techniques mentioned in section 3.3 are applied to the "real" training data for model training as a comparison. For OOD tasks, we evaluate trained models on stylized images, including "clippart", "infograph", "inpainting" and "sketch" styles in DomainNet. All model parameters remain the same as those in previous sections.

To improve the diversity of SD-generated images, we increase the variance of prompts for the stable diffusion model. We provide four prompt formats that are randomly picked for individual image-generating processes. Moreover, we dynamically adjust the generated images by randomly selecting and applying compressed resolutions within the ranges of 320-640 pixels for the first dimension and 240-720 pixels for the second dimension.

We use *Top-1* accuracy and Expected Calibration Error (ECE) as evaluation metrics for both ID and OOD tasks. ECE is widely used for model uncertainty evaluation. It quantifies how well a model's predicted probabilities (confidence) align with its actual outcomes (accuracy) (Guo et al., 2017). The smaller ECE is, the better the model is calibrated. Synthetic training datasets are created by replacing a certain proportion (15%, 25%, and 35%) of real images with SD-generated images, thereby the total training data size is kept the same as the baseline and other data augmentation methods.

| Data | ID | | OOD | | | | | | | | OOD Avg | |
|---|---|---|---|---|---|---|---|---|---|---|---|---|
| | | | Clipart | | Infograph | | Painting | | Sketch | | | |
| | Acc | ECE | Acc | ECE | Acc | ECE | Acc | ECE | Acc | ECE | Acc | ECE |
| baseline | 85.27 | 0.05 | 64.95 | 0.12 | 31.21 | 0.25 | 54.89 | 0.20 | 51.00 | 0.22 | 50.51 | 0.20 |
| auto | 85.03 | 0.04 | **67.99** | 0.10 | 31.93 | 0.10 | 57.74 | 0.18 | 55.83 | 0.18 | 53.37 | 0.14 |
| rand | 85.11 | 0.04 | 65.73 | 0.10 | 31.77 | 0.21 | **58.35** | 0.16 | 56.01 | 0.16 | 52.97 | 0.16 |
| augM | 84.48 | 0.04 | 67.10 | 0.10 | 31.69 | 0.21 | 55.98 | 0.18 | 54.51 | 0.18 | 52.32 | 0.17 |
| trAu | 84.30 | 0.03 | 65.93 | 0.08 | 31.12 | 0.20 | 55.31 | 0.16 | 54.17 | 0.17 | 51.63 | 0.15 |
| noise10 | 84.41 | 0.03 | 66.69 | 0.08 | 32.12 | 0.18 | 56.76 | 0.16 | 57.18 | 0.14 | 53.19 | 0.14 |
| noise30 | 84.45 | 0.03 | 67.59 | 0.07 | 32.68 | 0.16 | 56.76 | 0.15 | 58.12 | 0.14 | 53.79 | 0.13 |
| noise50 | 83.51 | 0.03 | 64.87 | 0.08 | 30.44 | 0.19 | 55.77 | 0.15 | 54.55 | 0.15 | 51.41 | 0.14 |
| auto-v1 | 83.44 | 0.03 | 65.53 | 0.07 | 31.59 | 0.15 | 55.93 | 0.14 | 55.73 | 0.14 | 52.20 | 0.13 |
| auto-v2 | 81.86 | 0.02 | 64.32 | 0.06 | 27.82 | 0.16 | 53.96 | 0.13 | 53.67 | 0.13 | 49.94 | 0.12 |
| auto-v3 | 75.89 | 0.03 | 55.37 | 0.06 | 23.71 | 0.15 | 49.37 | 0.13 | 46.68 | 0.13 | 43.78 | 0.12 |
| synthetic(0.15) | **85.85** | 0.05 | 66.91 | 0.11 | **33.00** | 0.19 | 56.65 | 0.15 | **59.97** | 0.15 | **54.13** | 0.15 |
| synthetic(0.25) | 85.11 | 0.05 | 67.21 | 0.07 | 32.28 | 0.20 | 53.84 | 0.17 | 54.98 | 0.12 | 52.08 | 0.14 |
| synthetic(0.35) | 84.58 | 0.05 | 64.59 | 0.14 | 30.58 | 0.22 | 54.75 | 0.15 | 51.07 | 0.17 | 50.25 | 0.17 |

Table 2: The *Top-1* accuracy and ECE evaluations of ID and OOD tasks for the CLIP (VIT-32 backbone) trained and tested on DomainNet. Both traditional data augmentation methods and the synthetic data method can result in better OOD performance.

**Result.** Table 2 shows that both traditional data augmentation methods and the synthetic data method can result in better OOD performance, i.e., higher *top-1* accuracy and lower ECE on average compared with the baseline. Extremely augmented data such as *auto-v2* and *auto-v3* may harm the model's learning ability. Figure 5 shows that Vendi Score has a strong correlation with both ID and OOD average performance. It also shows that, on DomainNet, data augmentation can increase the diversity score at the pixel level but fails at the embedding level. Similarly, infusing a larger proportion of synthetic data into the training dataset may not bring additional benefits; instead, it can decrease the model's performance in terms of classification accuracy. A recent publication (Alemohammad et al., 2023) describes a similar phenomenon called *model collapses*, which is caused by excessively using synthetic data to retrain gener-

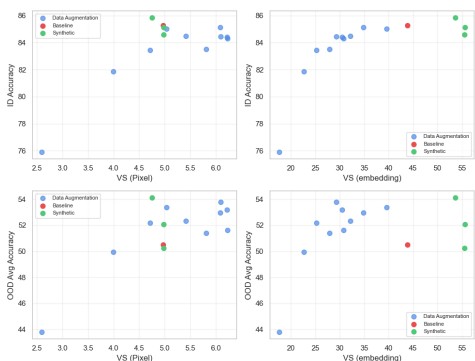

Figure 5: Correlation between Vendi Score and ID/OOD Accuracy. Vendi Score has a strong correlation with both ID and OOD average performance on DomainNet.

ative models. This may be partially due to the lack of diversity of synthetic data itself. The more iterations of retraining, the less diverse the generated synthetic data will be. Our findings demonstrate the importance of data diversity in enhancing machine learning capability.

## 4 RELATED WORK

Data diversity and applications of synthetic data have been getting more and more attention from the machine learning research community in recent years. Authors in the paper (Lopes et al., 2020) explore the role of diversity in data augmentation and propose two diversity definitions. One definition is model-based, or based on the performance of classification, and the other definition is input image data-based, by comparing the conditional entropy of clean data distribution and the augmented data distribution. They conclude that the diversity produced by data augmentation is able to improve model performance. However, it falls short of explaining the underlying mechanisms of how data diversity affects the ML model. The study from He et al. (2022) evaluates the use of synthetic data, which are generated by test-to-image models such as GLIDE, for image recognition tasks. It concludes that models trained solely on synthetic data do not outperform those trained on real data. Following this work, we try to explain why using synthetic data alone to train a model cannot beat what real data does. Similarly, model collapses have been observed by the authors of Alemoham-

mad et al. (2023). They state that the self-consuming loop – that is, iteratively using synthetic data from generative models as their own training data – can significantly harm the quality of generated output. They suggest adding fixed real data to break the loop and to improve the quality of generated images. In our experiments, we only use synthetic data to replace a small portion of real data so as to increase overall data diversity and improve model performance on both ID and OOD tasks.

The spectral analysis of weight metrics has been applied to various scenarios. Martin & Mahoney (2021) found deep neural networks naturally regularize themselves by shaping their weight matrices, for which they proposed a Heavy-Tailed Self-Regularization (HTSR) theory. Their follow-up work (Martin et al., 2021) used the power-law fitting on weight matrices' spectra to predict a model's quality and generalization ability without needing training or testing data, thus purely focusing on the intrinsic properties of the model's weights. Yang et al. (2023) applied the HTSR theory in the natural language processing area. They proposed a KS distance-based estimation method to make a better power law fitting, making the exponent $\alpha$ better capture the shape of the tail distribution of ESD. On the other hand, the significance of weight matrices has been explored in other recent studies. Wortsman et al. (2022) demonstrated that merging weights from different fine-tuning models can improve overall model performance and highlighted the impact of weight manipulation on optimizing model behavior without retraining. All of these prior studies motivated us to investigate the impact of data diversity from the model's weight matrices perspective.

## 5 DISCUSSION

Model generalizability is an open challenge for deploying machine learning and deep learning models, as it directly impacts their adaptability and reliability in real-world applications. Regularization techniques, such as dropout and weight decay, are typically used to reduce overfitting and improve model robustness. Data augmentation, an empirical method that increases the diversity of training data, particularly in situations where training data is scarce, could achieve similar effects. However, the underlying mechanisms of how and why data augmentation enhances model performance remain an open question in theory and practice. With advancements in generative AI, the concept of *data diversity* has received significant attention with respect to both training data and model outputs. Diverse training data enables large pre-trained models to learn a wide range of meaningful features without overfitting to noise, while ensuring diversity in generated outputs is crucial for their use as effective training data in downstream tasks. This motivates our investigation into the role of data diversity in the learning process of a model. Specifically, we aim to understand how data diversity influences the weight landscape and provide insights into why data augmentation can lead to more robust models. To achieve this, we used Random Matrix Theory (RMT) to analyze spectral patterns in the weight matrices of neural networks, comparing the effects of dropout, weight decay, and various data augmentation techniques. Additionally, we quantified the diversity of training sets using the Vendi score, which measures data variability and helps assess how training diversity contributes to model generalization in our framework.

We observed that dropout and data augmentation exhibit similarities in how they affect the weight space of neural networks. Both methods reduce the overall magnitude of the weight matrix and induce similar changes in the spectral metrics of the empirical spectral density (ESD) on the same dataset. This suggests that they influence weight formation in a comparable way: dropout forces the model to be less reliant on specific weight correlations, resulting in a less concentrated ESD, while the rich features in a diverse dataset encourage the model to focus on a broader range of information during training, leading to a more balanced weight landscape. In contrast, weight decay, another regularization method, also reduces the overall size of the weight matrix but impacts the shape of the ESD differently, leading to distinct regularization effects.

We also explored the use of synthetic data generated by generative models as an augmentation strategy to improve model performance. While previous studies have not thoroughly examined its effectiveness, we compared this new approach with traditional data augmentation in both in-distribution (ID) and out-of-distribution (OOD) tasks. Our results show that incorporating a small amount of synthetic data enhances average model performance on both ID and OOD tasks, primarily due to the increased diversity provided by the synthetic data. However, there is also a risk of model collapse when relying too heavily on synthetic data, which occurs when the model starts overfitting to synthetic patterns that lack the variability and quality of real-world data. This over-reliance can

reduce generalizability, cause biased learning, and diminish the model's ability to recognize a broad range of real-world features, ultimately making the model less robust and effective in practical applications. Therefore, a careful balance between real and synthetic data is still necessary to prevent model collapse and prevent overfitting.

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

# A    APPENDIX

We present more scale-shape plots here:

(a) Frobenius norm vs. alpha on CIFAR-10

(b) Frobenius norm vs. alpha on CIFAR-100

(c) Frobenius norm vs. entropy on Imagenette

(d) Frobenius norm vs. alpha on Flowers102

(e) Frobenius norm vs. entropy on Stanford Cars

(f) Frobenius norm vs. entropy on DomaineNet (Real)

Figure 6: Scale-shape plots of ESDs of the classifier layers on CIFAR-10/100, Imagenette, Flower102, StandfordCars, DomainNet(Real). The results are marked with square and diamond shapes for dropout and weight decay regularization techniques, respectively. The size of markers represents the magnitude of the dropout rate and weight decay values. The results above are based on the ResNet50 backbone. Because of limitations of computing resources, CIFAR-10/100 and DomainNet(Real) refer to a subset that samples 5000 images for fine-tuning.

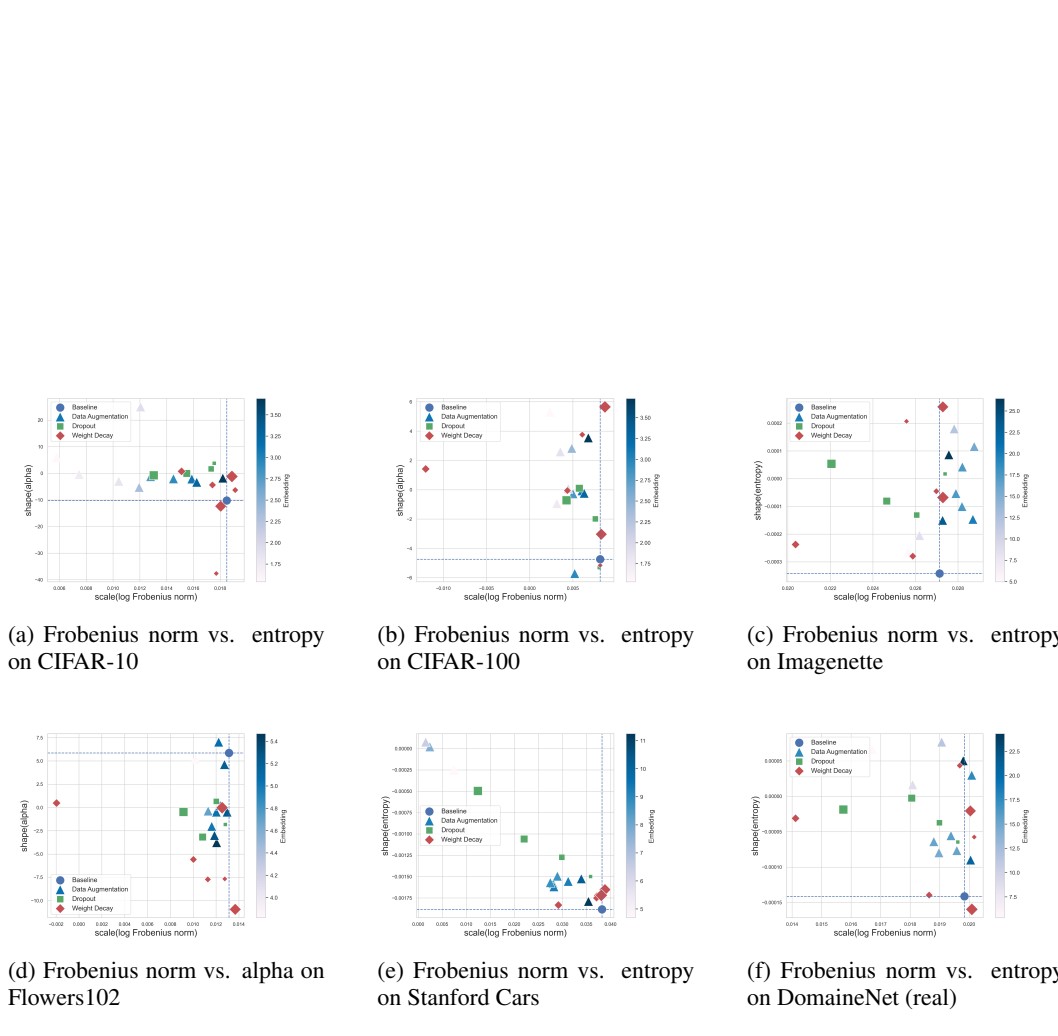

(a) Frobenius norm vs. entropy on CIFAR-10

(b) Frobenius norm vs. entropy on CIFAR-100

(c) Frobenius norm vs. entropy on Imagenette

(d) Frobenius norm vs. alpha on Flowers102

(e) Frobenius norm vs. entropy on Stanford Cars

(f) Frobenius norm vs. entropy on DomaineNet (real)

Figure 7: Scale-shape plots of ESDs of the classifier layers on CIFAR-10/100, Imagenette, Flower102, StandfordCars, DomainNet(Real). The results are marked with triangle, square, and diamond shapes for data augmentation, dropout, and weight decay, respectively. The color intensity of triangles represents the magnitude of embedding-based diversity scores. The results above are based on the ResNet50 backbone. Because of limitations of computing resources, CIFAR-10/100 and DomainNet(Real) refer to a subset that samples 5000 images for fine-tuning.

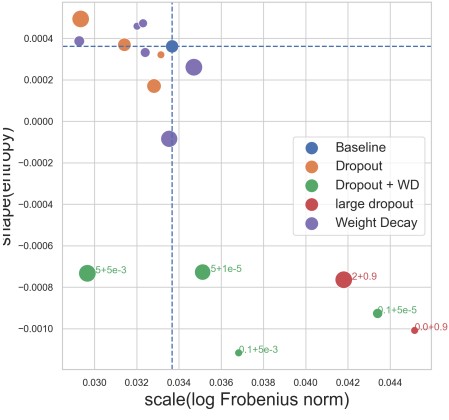

(a) Frobenius norm vs. entropy on Imagenette (ViT)

(b) Frobenius norm vs. entropy on Flowers102 (ViT)

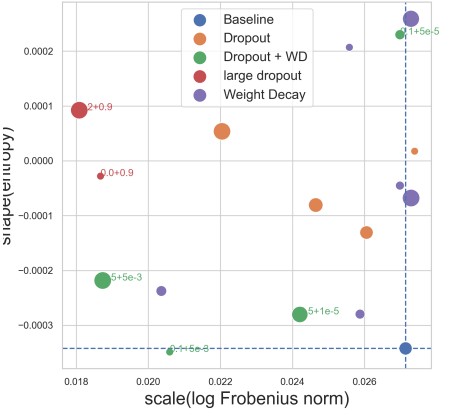

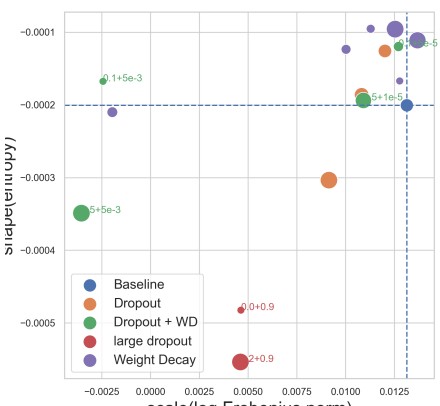

(c) Frobenius norm vs. entropy on Imagenette (ResNet50)

(d) Frobenius norm vs. entropy on Flowers102 (ResNet50)

Figure 8: Scale-shape plots of ESDs of the classifier layers on Imagenette and Flower102. The results are marked with square and diamond shapes for dropout and weight decay regularization techniques, respectively. The size of markers represents the magnitude of the dropout rate and weight decay values. We also refer (Zhang & Bottou, 2024) to apply a very large dropout (0.9 at the last layer), as well as combinations with dropout(0.1, 0.5) and weight decay (1e-5, 5e-3). Results show us combinations with dropout and weight decay, as well as the very large dropout can make more intense changes in both scale and shape side.

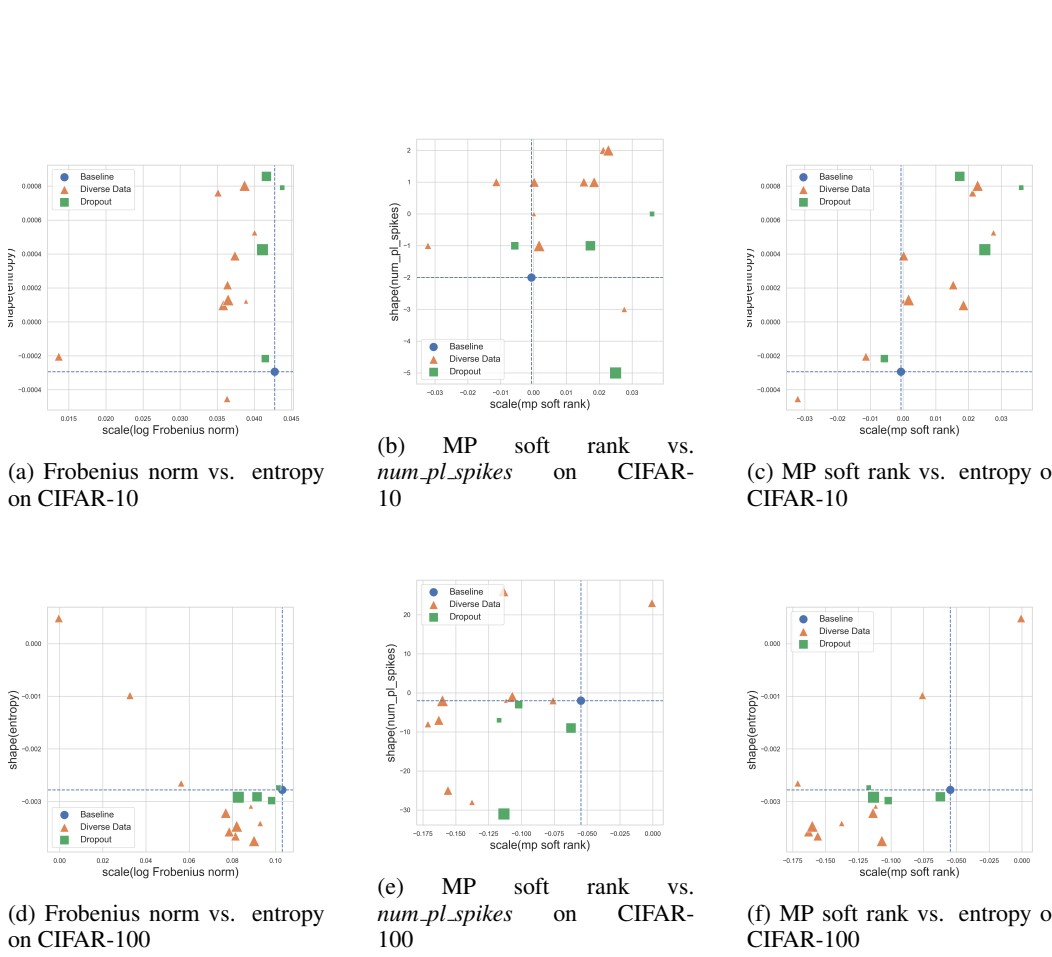

Figure 9: Various scale-shape plots of ESDs of the classification layer on CIFAR-10 and CIFAR-100. The results are marked with triangle and square shapes for data augmentation and dropout, respectively. **The size of markers represents the magnitude of the dropout rate and embedding-based Vendi score. More metrics from both scale and shape sides to compare dropout with data augmentations. These plots strengthen our conclusion about the similarity between dropout and diverse data in section 3.**

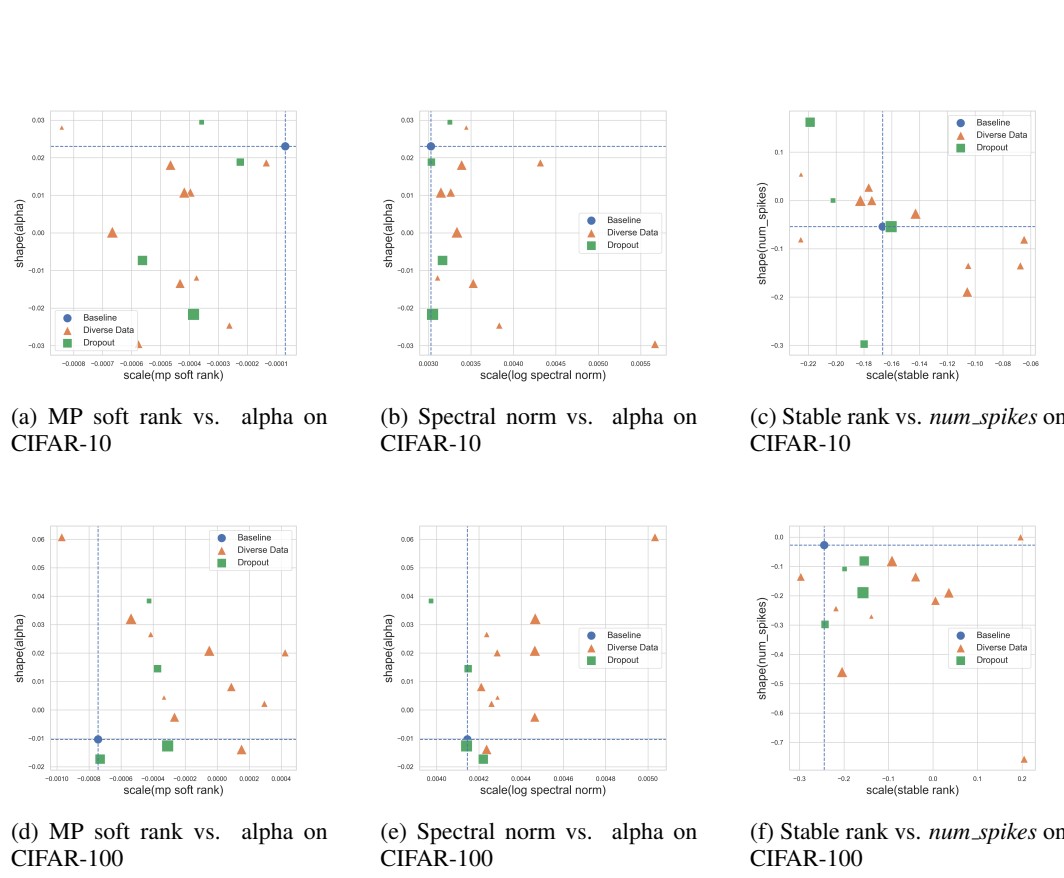

(a) MP soft rank vs. alpha on CIFAR-10

(b) Spectral norm vs. alpha on CIFAR-10

(c) Stable rank vs. *num_spikes* on CIFAR-10

(d) MP soft rank vs. alpha on CIFAR-100

(e) Spectral norm vs. alpha on CIFAR-100

(f) Stable rank vs. *num_spikes* on CIFAR-100

Figure 10: Various scale-shape plots of ESDs of all vision layers on CIFAR-10 and CIFAR-100. **We calculated the average changes through all layers in the vision encoder. The results are marked with triangle and square shapes for data augmentation and dropout, respectively. The size of markers represents the magnitude of the dropout rate and embedding-based Vendi score. In addition to the classification layer, the average changes of all vision encoder layers in CLIP exhibit similarities between dropout and data augmentation in both shape and scale. Even though the way they shape the distribution of eigenvalues is similar, diverse data changes of scale side to a larger value than dropout. Also, their changes compared with the baseline have a similar trend.**

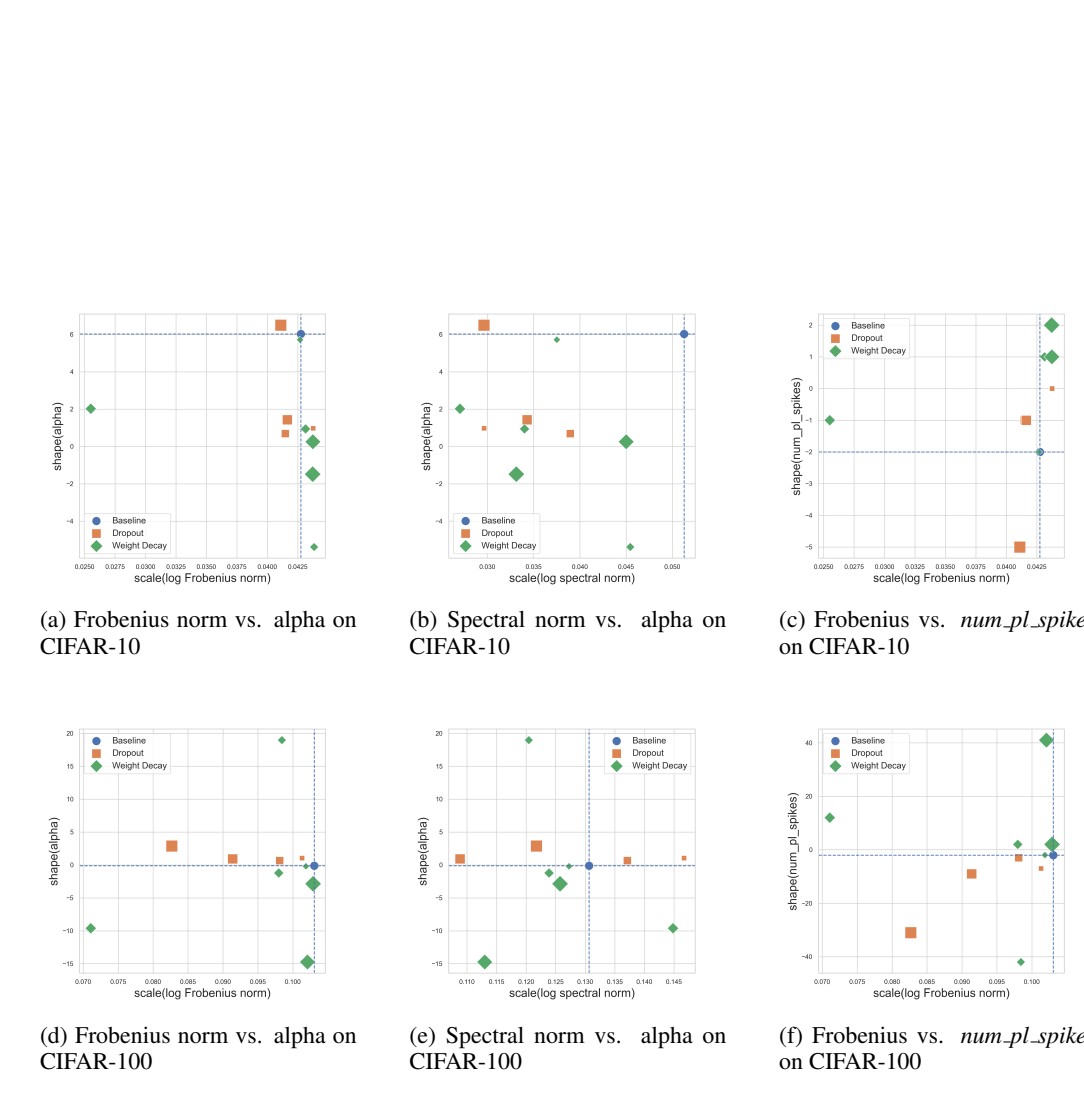

(a) Frobenius norm vs. alpha on CIFAR-10

(b) Spectral norm vs. alpha on CIFAR-10

(c) Frobenius vs. *num_pl_spikes* on CIFAR-10

(d) Frobenius norm vs. alpha on CIFAR-100

(e) Spectral norm vs. alpha on CIFAR-100

(f) Frobenius vs. *num_pl_spikes* on CIFAR-100

Figure 11: Various scale-shape plots of ESDs of the classifier layers on CIFAR-10 and CIFAR-100. The results are marked with square and diamond shapes for dropout and weight decay regularization techniques, respectively. The size of markers represents the magnitude of the dropout rate and weight decay values.

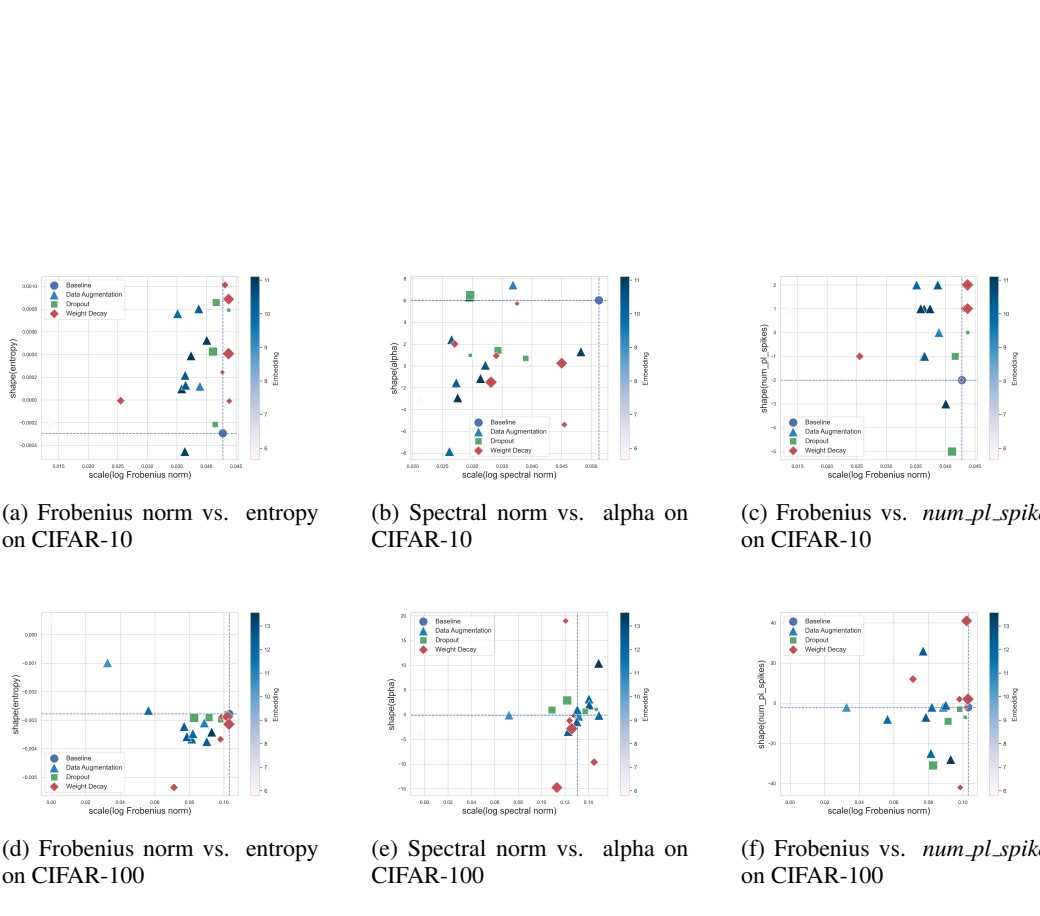

(a) Frobenius norm vs. entropy on CIFAR-10

(b) Spectral norm vs. alpha on CIFAR-10

(c) Frobenius vs. *num_pl_spikes* on CIFAR-10

(d) Frobenius norm vs. entropy on CIFAR-100

(e) Spectral norm vs. alpha on CIFAR-100

(f) Frobenius vs. *num_pl_spikes* on CIFAR-100

Figure 12: Various scale-shape plots of ESDs of the classifier layers on CIFAR-10 and CIFAR-100. The results are marked with triangle, square, and diamond shapes for data augmentation, dropout, and weight decay, respectively. The color intensity of triangles represents the magnitude of embedding-based diversity scores.

