# OpenReview forum: "How Does Data Diversity Shape The Weight Landscape of Neural Networks?"
_ICLR.cc/2025/Conference — Submitted to ICLR 2025_

### Official Review · Reviewer_cKCt · 2024-10-21

**Soundness:** 3
**Presentation:** 3
**Contribution:** 3
**Rating:** 6
**Confidence:** 3

**Summary:**

The paper provides a new insight into data diversity shape and its relationship with the weight landscape of neural networks. Furthermore, it investigates how data diversity can influence the weight matrices of neural networks. It focuses on comparing the impact of traditional regularization (dropout, weight decay) and data augmentation on neural networks. The authors used the Vendi Score, which measures the diversity in datasets, to quantify how diverse datasets, including synthetic data from generative models, affect model generalization. It finds that synthetic data can enhance model performance when combined with real data but can also cause model collapse when overused.

**Strengths:**

In general, the topic is interesting, and the paper is well-written. It provides good insights into the impact of diversity and synthetic data for better generalization with respect to the neural network landscape. Furthermore, the diversity per class for different augmentation strategies is motivating.

**Weaknesses:**

The work seems good, but it lacks more baselines and more motivation about its importance. Data diversity and generalization is a hot topic, as we can see in new works such as Hemmat, Reyhane Askari, et al. "Improving Geo-diversity of Generated Images with Contextualized Vendi Score Guidance." arXiv preprint arXiv:2406.04551 (2024). So, having a deep analysis of it is important. Furthermore, the work says that synthetic data can hurt the generalization data, but I didn't see any in-depth analysis of it, such as a graphic describing the amount of synthetic data vs. performance or diversity of the final model.


1 - For instance, the title of the work is "How does data diversity shape the weight landscape of neural networks?" but the experiments are done only with the CLIP VIT model; if possible, it would be good to have additional experiments with other models/backbone and also another dataset such as ImageNet (if not possible due to hardware constraints, consider using subsets of it such as imagenette), this would make the work more robust and grounded in better-evaluating settings.

2 - Figure 1 only brings cifar10, but this analysis is nice to have for other datasets as well (if not on the main paper, you can add it to the supplementary material).

3 - There is no visualization of the loss landscape; I think this is an important opportunity to show the behavior of data augmentation or synthetic data in the loss landscape.

4 - Other baselines such as Fine-tuning with Very Large Dropout (Zhang, Jianyu, and Léon Bottou), Dropout+Weight Decay (a combination of the two baselines of Fig. 3) would be interesting to have.

**Questions:**

For me the authors can work on the Weaknesses list described. I think the work has its merits, but the authors need to address some of the points mentioned in the Weaknesses section.

Some important points for the authors:

1. Does the pre-train zero-shot model affect the diversity or data augmentation used? Or can the landscape change if you start from a model from scratch? This can be a good analysis with a small model such as resnet18 or resnet34 (I am not saying to test it with CLIP, but this could be an interesting point). Additionally, why only choose CLIP VIT specifically, and do you believe that your findings would generalize to other architectures? How the pre-training model could interact with data diversity effects, and do you expect different results for models trained from scratch versus fine-tuned models.

2. Do you think that loss landscape visualization would complement your current analysis?

3. "Therefore, a careful balance between real and synthetic data is still necessary to prevent model collapse and prevent overfitting." Could we use a diversity metric inside the training to guide the augmentation needed or even to balance the amount of synthetic vs. real data? If so, do you think that the model would collapse or overfit the synthetic data? Discuss the potential challenges and benefits of incorporating a diversity metric into the training process,

The above questions can be used to improve some insights and analysis of the work. Furthermore, I didn't see anything about open-source code or reproducibility, which can be important for the scientific community.

I am happy to see the rebuttal phase and hope that the authors can do a good job of improving the points mentioned.

---

> ### Author Response · Authors · 2024-11-23
>
> We appreciate reviewer cKCt's valuable comments/suggestions. We provide our response below:
>
> > Data diversity and generalization is a hot topic, as we can see in new works such as Hemmat, Reyhane Askari, et al. "Improving Geo-diversity of Generated Images with Contextualized Vendi Score Guidance." arXiv preprint arXiv:2406.04551 (2024). So, having a deep analysis of it is important.
>
> Thanks for your comment. In this paper, our theoretical analysis relies on the Random Matrix Theory, which says well fine-tuned models have unique patterns in the weight matrices. We introduce a quant-chart visualization analysis approach from the scale and shape perspective to compare different characteristics among various regularization and data augmentation techniques.  We offer an intuitive understanding of the observed effects of data diversity in the paper (Lines 372-376), as compared with those of regularization methods. In our future research, we will employ the Bayesian framework to dip into the reason why data augmentation can lead to a better generalization and provide a more solid theoretical study.
>
> > Furthermore, the work says that synthetic data can hurt the generalization data, but I didn't see any in-depth analysis of it, such as a graphic describing the amount of synthetic data vs. performance or diversity of the final model.
>
> Thanks for the comments. We observe the model collapse by comparing the performance when adding more synthetic data in fine-tuning. Adding 25% and 35% synthetic data makes the model performance worse than 15% synthetic data inclusion. Analyzing model collapse in detail is not the scope of this paper. In our future research, we will study the process of model collapse and uncover the reason this phenomenon happens, as well as methods to mitigate it.
>
> If the reviewer is interested in the model collapse, here are some newly published papers to read:
> - Dohmatob, Elvis, et al. "A tale of tails: Model collapse as a change of scaling laws." arXiv preprint arXiv:2402.07043 (2024).
> - Dohmatob, Elvis, Yunzhen Feng, and Julia Kempe. "Model collapse demystified: The case of regression." arXiv preprint arXiv:2402.07712 (2024).
> - Feng, Yunzhen, et al. "Beyond Model Collapse: Scaling Up with Synthesized Data Requires Reinforcement." arXiv preprint arXiv:2406.07515 (2024).
>
> We have a description of the amount of synthetic data, 0.15, 0,.25, and 0.35 a different proportion of synthetic data added to the real data while keeping the total amount of data the same, which means a large proportion indicates more synthetic data is included in the fine-tuning. (Lines 429-431). We updated a figure connected the VS with ID and OOD performance in our synthetic data experiment, which could address your concerns and provide some insights, please check our revised manuscript (blue texts).
>
> > 1 - For instance, the title of the work is "How does data diversity shape the weight landscape of neural networks?" but the experiments are done only with the CLIP VIT model; if possible, it would be good to have additional experiments with other models/backbone and also another dataset such as ImageNet (if not possible due to hardware constraints, consider using subsets of it such as imagenette), this would make the work more robust and grounded in better-evaluating settings.
>
> We appreciate the recommendations from the reviewer. We added some other experiments including Imagenette and three other high-resolution datasets, as well as an additional backbone ResNet50. Please check the result in section 3 and the Appendix in the revised manuscript. In short, our observations still hold on ResNet50 for Clip.
>
> > 2 - Figure 1 only brings cifar10, but this analysis is nice to have for other datasets as well (if not on the main paper, you can add it to the supplementary material).
>
> We added experiments on high-resolution datasets and additional backbone ResNet50. Please check the results of section 3 and the Appendix in the revised manuscript.
>
> > 3 - There is no visualization of the loss landscape; I think this is an important opportunity to show the behavior of data augmentation or synthetic data in the loss landscape. And Question: Do you think that loss landscape visualization would complement your current analysis?
>
> We appreciate the reviewer raising this suggestion. We recognize the importance of loss landscapes in understanding neural network optimization. We also think loss landscape visualization can provide interesting insights into understanding different regularization and data augmentations.
>
> However, the idea of our paper is not to discover the dynamics of the loss surface during fine-tuning. Our paper serves as post and structural analysis when fine-tuning is done, where we can obtain insights from the weights of well-tuned models.

---

> > ### Author Response · Authors · 2024-11-23
> >
> > > 4 - Other baselines such as Fine-tuning with Very Large Dropout (Zhang, Jianyu, and Léon Bottou), Dropout+Weight Decay (a combination of the two baselines of Fig. 3) would be interesting to have.
> >
> > We added these experiments and compared weight decay + dropout and very large dropout. The results are in Figure 8 in the appendix. For Dropout + Weight Decay, we choose dropout rates 0.1, and 0.5, weight decay 1e-5 and 5e-3. Four combinations of them are tested. For a very large dropout, we have two settings: one is 0.0 dropout in hidden layers and 0.9 for the last layer, while the other is 0.2 dropout in hidden layers and 0.9 for the last layer. Results show us combinations with dropout and weight decay, as well as the very large dropout, can make more intense changes in both the scale and shape side. Their positions are highlighted in Figure 8.
> >
> > > Does the pre-train zero-shot model affect the diversity or data augmentation used? Or can the landscape change if you start from a model from scratch? This can be a good analysis with a small model such as resnet18 or resnet34 (I am not saying to test it with CLIP, but this could be an interesting point). Additionally, why only choose CLIP VIT specifically, and do you believe that your findings would generalize to other architectures? How the pre-training model could interact with data diversity effects, and do you expect different results for models trained from scratch versus fine-tuned models.
> >
> > Thanks for raising the questions. Without fine-tuning or model training, zero-shot prediction would not affect the diversity or data augmentation used too much, or it’s challenging to measure if the effect exists.
> >
> > If training a model from scratch, we expect a similar but may not be exactly the same effect. Specifically, we expect the shape metrics to stay the same with the results of fine-tuning, and shape metrics may vary based on different datasets used in the training process.  Actually, in our fine-tuning, we unfreeze all layers in the vision encoder, whose weights can be fully updated.  In the appendix, there are more figures relating to other layers.
> >
> > We have ResNet50 as a new backbone, please check the results in the Appendix.
> >
> > > "Therefore, a careful balance between real and synthetic data is still necessary to prevent model collapse and prevent overfitting." Could we use a diversity metric inside the training to guide the augmentation needed or even to balance the amount of synthetic vs. real data? If so, do you think that the model would collapse or overfit the synthetic data? Discuss the potential challenges and benefits of incorporating a diversity metric into the training process
> >
> > Thanks for raising this concern. We believe diversity measurement can be seen as one of the important considerations during the training process. We observe a positive correlation between VS (See Figure 5 in the revised manuscript), informing us that more diverse data can achieve better generalization performance. However, the data diversity caused by data augmentation is at the pixel level while synthetic data can result in embedding level diversity increase by involving new information, which could be the reason it has better both ID and OOD performance.
> >
> > Through our empirical experiment and existing literature, we believe the model will collapse when more and more low-quality synthetic data is included in the training process because of the distribution shift. We think that a good balance would be beneficial as well as an efficient synthetic data selection criteria, where we could know what kind of synthetic data is necessary.  We will add this observation and discussion in the final version.
> >
> > > I didn't see anything about open-source code or reproducibility, which can be important for the scientific community.
> >
> > Thanks for raising this concern, we will release our code when the paper is about to be published.

---

> > > ### Comment · Reviewer_cKCt · 2024-11-25
> > >
> > > Hello, authors; after reading the reviews carefully, I think that my questions and concerns were answered. So, I decided to change the contributions from 2 to 3, and the scores from "5: marginally below the acceptance threshold" to "6: marginally above the acceptance threshold".
> > >
> > > Even though I am comfortable with the answers of the authors, I still think that there is some good analysis that could improve the work quality, such as the ones that I mentioned before.
> > >
> > > Thanks for the work done and the clarification during the rebuttal phase.

---

> > > > ### Author Response · Authors · 2024-11-27
> > > >
> > > > Thank you reviewer cKCt. We appreciate your valuable recommendations to improve our work. We will carefully consider including them in the future work. Thanks once again for taking the time to review our work.

---

### Official Review · Reviewer_DNCf · 2024-11-03

**Soundness:** 3
**Presentation:** 3
**Contribution:** 3
**Rating:** 5
**Confidence:** 3

**Summary:**

This paper explores the effects of various regularization and augmentation techniques on the parameter space of neural networks, with a particular focus on the weight landscape in transfer learning contexts. Specifically, it employs Random Matrix Theory to examine the distribution of eigenvalues between pretrained and finetuned models that utilize these techniques. Additionally, the paper conducts comparative experiments across diverse datasets to further investigate these effects.

**Strengths:**

This paper leverages Random Matrix Theory to analyze the impact of augmentation and regularization techniques, providing a valuable perspective for examining more complex methods.
The paper puts forward several arguments, notably that diverse data can enhance model performance.
The study includes multiple experiments designed to investigate the effects of various regularization and augmentation strategies.

**Weaknesses:**

While this paper effectively explores the impact of various regularization techniques and provides some explanations, it primarily resembles an experimental report. I am curious whether the findings from these experiments could be utilized to optimize the application of regularization or data augmentation techniques.
Moreover, the focus of the paper is predominantly on experimental validation, and the use of mainly the CIFAR dataset might not be sufficiently representative. It would be beneficial to include additional datasets to strengthen the validity of the results.
Regarding the explanations provided for the findings, could the authors offer some theoretical analysis to elucidate why these phenomena occur? This would enhance the depth of the paper and provide a stronger theoretical foundation for the observed effects.

**Questions:**

Please refer to weakness

---

> ### Author Response · Authors · 2024-11-23
>
> We thank the reviewer DNCf's time and feedback. We provide our response below:
>
> > While this paper effectively explores the impact of various regularization techniques and provides some explanations, it primarily resembles an experimental report.
>
> Thanks for the comment. However, we believe this paper is much more than an experimental report. Our analytical approach has a rigorous theoretical foundation. The goal of this paper is to explore weight matrix changes caused by different regularization and data diversity techniques via Random Matrix Theory. We believe using this technique to tackle new problems carries an important contribution as well. Moreover, even though we didn’t particularly highlight the quant-chart visualization analysis approach in our paper, we do believe it’s original too.
>
> In practice, data augmentation has been seen as a very efficient approach to enhancing model generality. Our work provides a math-based explanation of this phenomenon via the model weight landscape characterization. We compare the effects of training data diversity on the weight landscape with the effects of model regularization such as dropout and weight decay. Furthermore, our paper also discusses a reason for using synthetic data – to increase data diversity.
>
> > I am curious whether the findings from these experiments could be utilized to optimize the application of regularization or data augmentation techniques.
>
> Thanks for pointing this out. We observed the Vendi Score has a positive correlation with both ID and OOD accuracy on DomainNet. We also observed that not all data augmentations can make Vendi Score increase, especially for embedding-based VS. Therefore, we believe that these findings can offer insights for further optimizing the application of regularization or data augmentation techniques, or even directing the development of new augmentation techniques.
>
> > Moreover, the focus of the paper is predominantly on experimental validation, and the use of mainly the CIFAR dataset might not be sufficiently representative. It would be beneficial to include additional datasets to strengthen the validity of the results.
>
> Thanks for raising this concern. We added several more datasets and used ResNet50 as the additional backbone for the clip model. Please see our revised manuscript. Additional datasets are included in the main body of the paper and the results for ResNet50 are included in the appendix.
>
> > Regarding the explanations provided for the findings, could the authors offer some theoretical analysis to elucidate why these phenomena occur? This would enhance the depth of the paper and provide a stronger theoretical foundation for the observed effects.
>
> Thanks for your comment. In this paper, our theoretical analysis relies on the Random Matrix Theory, which says well fine-tuned models have unique patterns in the weight matrices. We introduce a quant-chart visualization analysis approach from the scale and shape perspective to compare different characteristics among various regularization and data augmentation techniques.  We offer an intuitive understanding of the observed effects of data diversity in the paper (Lines 372-376), as compared with those of regularization methods. In our future research, we will employ the Bayesian framework to dip into the reason why data augmentation can lead to a better generalization and provide a more solid theoretical study.

---

> > ### Author Response · Authors · 2024-11-27
> >
> > Dear reviewer DNCf,
> >
> > Thank you for taking the time to review our work and provide insightful feedback.
> >
> > We have thoroughly addressed each of your comments and included detailed responses to clarify and improve the manuscript. We greatly value the opportunity to engage in a constructive discussion. If you have any additional questions or require further clarification, please don’t hesitate to leave comments. We remain available to address any further questions you may have.
> >
> > We hope our responses align with your expectations and would kindly invite you to reconsider your evaluation score.
> >
> > The Authors

---

### Official Review · Reviewer_777u · 2024-11-03

**Soundness:** 2
**Presentation:** 2
**Contribution:** 1
**Rating:** 3
**Confidence:** 3

**Summary:**

The submission makes use of perspectives about how spectral analysis of weight matrices in neural networks relate to its regularization properties developed in “Traditional and heavy-tailed self regularization in neural network models”, Martin and Mahoney, 2019.  The key assumption is that similar deviations in the eigenspectrum from that predicted by random matrix theory signify equivalent generalization properties.

Under this assumption, the submission explores the spectral effect data diversity has on the weight matrices of a transformer-based neural network being fine-tuned with different levels and varieties of data diversity, relating the changes to those induced by traditional regularization techniques such as dropout and weight decay.

Experiments are performed by fine-tuning a CLIP vision encoder on CIFAR 10 and 100. Results suggest that data augmentations and some amount of synthetic data inclusion have similar effects on the empirical eigenspectrum as dropout while differing in some aspects from weight decay.

**Strengths:**

Originality: While the method for analysis is borrowed, the particular application in the context of comparing data augmentation approaches with other regularizers is new, to my knowledge.

Clarity: The submission is easy to read, and the presentation is well-organized.

Quality and significance: The analysis is intriguing, and suggestive of further explorations.

**Weaknesses:**

The submission’s goal of using mathematical tools to inspect similarities between data augmentations and model-parameter based regularization strategies is intriguing. However, this goal is not adequately explored for the results to be considered informative enough to be interesting or actionable. Only one base model is used, and the choice of two small-scale image datasets is somewhat narrow.

The synthetic data experiments seem a little out of place, it was not clear to me why they fit in this paper, and what the connection is to the analytic method that seemed to be the central focus of the submission. In my opinion, these two aspects can be separate drafts, with considerably more thorough experimentation in order to make compelling cases for both.

Some typos:

L203: “Since the…” —> “Due to the…”?

L210-211: “properties of the spectral” —> “properties of the spectrum” or “spectral properties of the weight matrices”?

**Questions:**

1. Is CIFAR a good choice for fine-tuning experiments on a pre-trained CLIP B/32 model? Aren’t these models typically trained on higher resolution images?

2. L151 says that “We also observe that pixel-wise diversity scores do not always match embedding-wise scores after applying data augmentation.” Is the base model trained with some of these data augmentations already, thus learning to be invariant to them?

---

> ### Author Response · Authors · 2024-11-23
>
> We thank the reviewer 777u for the insightful comments. We provide our response below:
>
> > The submission’s goal of using mathematical tools to inspect similarities between data augmentations and model-parameter based regularization strategies is intriguing. However, this goal is not adequately explored for the results to be considered informative enough to be interesting or actionable. Only one base model is used, and the choice of two small-scale image datasets is somewhat narrow.
>
> Thanks for pointing it out. We acknowledge the limitations of the original draft. We added more datasets and used ResNet50 as another backbone, in addition to ViT-32, for the clip model. Please check our revised manuscript. Additional datasets are included in the main body of the paper and the results for ResNet50 are included in the appendix.
>
> > The synthetic data experiments seem a little out of place, it was not clear to me why they fit in this paper, and what the connection is to the analytic method that seemed to be the central focus of the submission. In my opinion, these two aspects can be separate drafts, with considerably more thorough experimentation in order to make compelling cases for both.
>
> Thanks for the comment. However, we respectfully disagree with the statement that the synthetic data experiments seem out of place in this paper.
>
> Given the rising popularity of generative models, using synthetic data has become a popular approach to training or fine-tuning models, especially in data scarcity scenarios. On one hand, synthetic data can be considered as another data augmentation technique; on the other hand, because the generator is usually pre-trained on a large amount of data, such as LLMs and stable diffusion, the generated synthetic data doesn’t come from the same distribution of the real data, therefore they would bring more diversified training data into the training process (when real and synthetic data are combined). This new phenomenon is worth exploring.
>
> Moreover, existing literature has not sufficiently investigated why synthetic data works, e.g., He, Ruifei, et al (2022). Our paper provides a reason from the training data diversity perspective. We added a new figure in the synthetic data experiment section. Please see the updated manuscript, and let us know if it addressed your concern.
>
> Reference: He, Ruifei, et al. "Is synthetic data from generative models ready for image recognition?." arXiv preprint arXiv:2210.07574 (2022).
>
> > Some typos:
> L203: “Since the…” —> “Due to the…”?
> L210-211: “properties of the spectral” —> “properties of the spectrum” or “spectral properties of the weight matrices”?
>
> Thanks for pointing it out, we fixed it in the revised manuscript.
>
> > Questions: 1. Is CIFAR a good choice for fine-tuning experiments on a pre-trained CLIP B/32 model? Aren’t these models typically trained on higher resolution images?
>
> Thanks for raising this concern, we added experiments on high-resolution datasets and additional backbone ResNet50. Please check the results of section 3 and the appendix in the revised manuscript.
>
> > 2. L151 says that “We also observe that pixel-wise diversity scores do not always match embedding-wise scores after applying data augmentation.” Is the base model trained with some of these data augmentations already, thus learning to be invariant to them?
>
> Even though data augmentation is a popular method for model training, there is no evidence that advanced data augmentation approaches have been applied in CLIP mode training. We refer to the original CLIP paper (Radford, Alec, et al. 2021), and only found this statement: “A random square crop from resized images is the only data augmentation used during training”.  The mismatch between pixel-wise diversity scores and embedding-wise diversity scores is caused by different approaches to similarity matrix calculation in the Vendi Score.
>
> Reference: Radford, Alec, et al. "Learning transferable visual models from natural language supervision." International conference on machine learning. PMLR, 2021.

---

> > ### Author Response · Authors · 2024-11-23
> >
> > Finally, we would like to emphasize more on the originality and contributions of our work. We appreciate the reviewer's acknowledgment that we are the first to use a Random Matrix Theory to explain the effects of data diversity. We believe using the existing approach to tackle new problems carries an important contribution as well. Moreover, even though we didn’t particularly highlight the quant-chart visualization analysis approach in our paper, we do believe it’s original too. In practice, data augmentation has been seen as a very efficient approach to enhancing model generality. Our work provides a math-based explanation of this phenomenon via the model weight landscape characterization. We compare the effects of training data diversity on the weight landscape with the effects of model regularization such as dropout and weight decay. Furthermore, our paper also discusses a reason for using synthetic data – to increase data diversity.
> >
> > We hope our responses above answer your concerns.  Please let us know if you have any other concerns.

---

> > > ### Author Response · Authors · 2024-11-27
> > >
> > > Dear reviewer 777u,
> > >
> > > Thank you once again for taking the time to review our work.
> > >
> > > We have taken your comments into careful consideration and included detailed responses to clarify and improve the manuscript. We greatly value the opportunity to engage in a constructive discussion. If you have any additional questions or require further clarification, please don’t hesitate to leave comments. We remain available to address any further questions you may have.
> > >
> > > We hope our responses effectively address your concerns and would kindly encourage you to reconsider your evaluation score.
> > >
> > > The Authors

---

> ### Comment · Reviewer_777u · 2024-11-28
> **Follow-up to rebuttal**
>
> Thanks to the authors for their follow-up. At this time, I'm inclined to maintain my initial rating.
>
> * It's great that the new datasets an additional model architecture has been added. However, it is not very obvious to me that the same patterns are showing up across all the plots, suggesting we might need to think more carefully about the takeaways.
>
> * Showing a correlation between diversity in synthetic examples and generalization is certainly interesting, but it still feels like a different topic to me than the way the paper is set up ("How does data diversity affect the weight landscape of neural networks?").
>
> * Thanks for clarifying about augmentation not being applied in the base networks!

---

> > ### Author Response · Authors · 2024-11-29
> >
> > Thanks for your response. Let us address your concerns further.
> >
> > > It's great that the new datasets an additional model architecture has been added. However, it is not very obvious to me that the same patterns are showing up across all the plots, suggesting we might need to think more carefully about the takeaways.
> >
> > Except for (a) and (c) in Figures 3 and 4, all plots illustrate that different regularization methods and data diversity techniques shift the weights to the left of the baseline in the dimension of scale metric (x-axis). This indicates that the entire weight becomes smaller compared with the baseline. Let’s examine these two exceptions: the commonality between CIFAR-10 (a) and Imagenette (c) is that both have 10 classes, whereas the other datasets contain between 100 and 196 classes. Therefore, the patterns are evident and consistent. Weight changes differ between datasets with fewer classes and those with more classes, but similar datasets exhibit consistent patterns.
> >
> > The shape metric (y-axis) suggests that data diversity behaves more like dropout than weight decay. In Figures 3 (b) and (f), dropout and weight decay move in opposite directions along the y-axis, while Figures 4 (b) and (f) show that data diversity aligns with the same direction as dropout. Similarly, in Figures 3 (d) and (e) and Figures 4 (d) and (e), although dropout and weight decay fall within the same quadrant, data diversity is closer to dropout.
> >
> > Although comparing the effects of different data diversity techniques and regularizations within the same dataset is more meaningful, both comparisons across different datasets and within the same dataset provide evidence to support the conclusions of our paper.
> >
> > > Showing a correlation between diversity in synthetic examples and generalization is certainly interesting, but it still feels like a different topic to me than the way the paper is set up ("How does data diversity affect the weight landscape of neural networks?").
> >
> > We disagree with your opinion regarding synthetic data. The title of our paper is “How does **data diversity** shape the weight landscape of neural networks” rather than “How does **data augmentation** shape the weight landscape of neural networks”. Synthetic data represents a novel approach to increasing data diversity, and we demonstrate how it influences the Vendi Score, a measure of data diversity, in comparison to traditional data augmentation methods. By including this analysis, we highlight the advantages and disadvantages of synthetic data, which we believe is a natural and integral part of the paper.

---

### Official Review · Reviewer_UKMS · 2024-11-04

**Soundness:** 2
**Presentation:** 2
**Contribution:** 2
**Rating:** 3
**Confidence:** 4

**Summary:**

This submission examines how data diversity shapes the weight landscape of neural networks. To investigate this, the study explores how techniques such as dataset augmentation and regularization methods impact the parameter space of neural networks, focusing on transfer learning scenarios. Random Matrix Theory is applied to analyze the eigenvalue distributions of pre-trained models, fine-tuned using these techniques with varying levels of data diversity for the same downstream tasks. The main observation is that diverse data influences the weight landscape in a similar way to dropout. Additionally, synthetic data created by generative models can increase diversity and improve out-of-distribution generalization.

**Strengths:**

+ Studying the impact of data augmentation on the landscape of weight parameters is interesting, and the use of Random Matrix Theory is straightforward.
+ The observation that “dropout and data augmentation exhibit similarities in how they affect the weight space of neural networks” is also intriguing. This observation seems expected and reasonable.

+ The final disucssion part is good. Serveral good points are made in disucussing the impact of regularization methods and data augmentation

**Weaknesses:**

- **The main concern is that the analysis methodology is not convincing**. This submission states, “since the heavy-tailed nature of pre-trained models… we focus on the trend of how regularization and diverse data influence the weight spectrum.” This statement is unclear. The weight differences observed are between a pre-trained model and a fine-tuned model, which are expected to be naturally different due to the use of different training data and objectives. It’s unclear why this difference is a valid measure of the effect of each technique.
Second, the Vendi Score (VS) is used to measure the intensity of diversity, which is acceptable. However, using different spaces—specifically, the raw pixel space versus the feature space—yields different observations, as shown in Figure 1. How should this difference be interpreted? Additionally, why is CLIP used instead of Inception? Also, the definition of VS(K) is unclear. What does K represent?

- **The analysis lacks clarity.** The ESD is used to illustrate the effect of each technique in Figure 3, but what is the main point? It’s challenging to draw clear observations from this figure. Figure 4 raises the same question. Additionally, what is the purpose of reporting Table 2, which merely lists numbers without providing a clear takeaway? Moreover, The order of classes in Figure1 will have impact, but this submission does not consider this.


***---Post Rebuttal---***

Thank you for providing the rebuttal. However, my original concerns remain unaddressed, so I am maintaining my score of 3.

**Questions:**

Please clarify and improve the analysis methodology. Additionally, the results lack clarity and do not consistently demonstrate a clear observation. While the discussion section is interesting, the analysis does not effectively support the main points.

---

> ### Author Response · Authors · 2024-11-23
>
> We appreciate reviewer UKMS’s efforts in reviewing our paper and feedback. We provide our response below:
>
> > The weight differences observed are between a pre-trained model and a fine-tuned model, which are expected to be naturally different due to the use of different training data and objectives. It’s unclear why this difference is a valid measure of the effect of each technique.
>
> The difference before and after fine-tuning itself doesn’t tell us anything. Instead, what we did is to compare the differences $\Delta$M’s from various fine-tuning methods with the baseline difference $\Delta$M.  This is why we introduced quadrant chart visualization analysis and measured the difference from two perspectives – scale and shape.  We did observe that different regularizations exhibit different patterns on these charts (see Figures 3 and 4). If different techniques fall in the same quadrant, we consider they have the same effect on changing the weight space.
>
> >  Second, the Vendi Score (VS) is used to measure the intensity of diversity, which is acceptable. However, using different spaces—specifically, the raw pixel space versus the feature space—yields different observations, as shown in Figure 1. How should this difference be interpreted?
>
> To clarify, Vendi Score (VS) was not used to measure diversity in the feature space, but it was used to evaluate the diversity introduced by various data augmentation and synthetic data methods. This highlights that different approaches bring distinct levels of diversity to the input data. Figure 1 provides an overview of how VS operates and offers insights into the diversity effects of different data augmentation techniques.
>
> > Additionally, why is CLIP used instead of Inception?
>
> Our experiments in the following paper are conducted via the CLIP model. To ensure consistency in diversity measurements with fine-tuned models, we replaced the CLIP model preprocessor with Inception. Different embedding models would result in different VS results. However, we believe this does not impact any main conclusions of our study. If we use Inception V3 to fine-tune, we will use Inception embeddings; if CLIP, then we use CLIP embeddings.
>
> > Also, the definition of VS(K) is unclear. What does K represent?
>
> We have this explanation in the paper (lines 134 -135). K represents the similarity matrix. The computation of the VS score is based on a similarity matrix and there are two methods to calculate a similarity matrix: pixel-based and embedding-based.
>
> > The analysis lacks clarity. The ESD is used to illustrate the effect of each technique in Figure 3, but what is the main point? It’s challenging to draw clear observations from this figure. Figure 4 raises the same question. Additionally, what is the purpose of reporting Table 2, which merely lists numbers without providing a clear takeaway?
>
> For each subsection in section 3, we have written a **summary** (Line 310-314, Line 370-377, and Line 451-470) to explain the main point.  Additionally, we have expanded the description below each figure and provided comprehensive discussions in subsequent paragraphs. After discussion, we wrote a summary to conclude our findings. We hope the reviewer could check our methodology part and experiment discussion, as well as the summaries, again, and let us know if any further explanation is needed.
>
> > The order of classes in Figure1 will have impact, but this submission does not consider this.
>
> Figure 1 doesn’t show us the order of classes matters. It tells us that increasing the number of classes may bring a certain level of diversity. We consider the diversity brought by increasing the number of classes. Our experiment includes various classes from cifar10(10 classes), cifar100 (100 classes), Imagenette(10 classes), Flower102 (102 classes), a subset of DomainNet (142 classes) to StandfordCars (196 classes).
>
> We hope our responses above answer your concerns.

---

> > ### Author Response · Authors · 2024-11-27
> >
> > Hello reviewer UKMS, we saw your brief reply. However, we carefully reviewed your previous comments and provided detailed responses to each of your considerations; therefore, we are looking for a more constructive discussion. We hope you could leave official comments to specify your concerns that were not addressed.

---

### Meta-Review · Area_Chair_T93e · 2024-12-21

**Metareview:**

The paper proposes to use Random Matrix Theory to reason that data diversity is as important as dropout, specifically by examining how the weight landscape of the optimization changes. Reviewers appreciated the attempt, however they raise criticism on the evaluation and analysis, both in experiments and theory. This was also confirmed in the post-rebuttal discussion. While the content is interesting, the paper certainly needs one more round of rewriting before acceptance.

**Additional Comments On Reviewer Discussion:**

Reviewers confirmed that the paper lacks convincing analysis methodology, and robust analysis, while in the rebuttal it was commented that quite a few suggestions were ignored, including reviewers that initially had a positive outlook.
Even thought I was positive about the work, I agree with other reviewers and I think that the paper is not ready for the conference and the authors ignored some of my suggestions as well. Thus, I agree with others about the recommendation to reject it.

Best Regards, Reviewer cKCt

---

### Decision · Program_Chairs · 2025-01-22

Reject